# A Bayesian game approach for node-based attribution defense against asymmetric information attacks in IoT networks

**Jun Chen**[1], **Xin Sun**[1], **Wen Tian**[2], **Guangjie Liu**[2]*

**1** School of Automation, Nanjing University of Science and Technology, Nanjing, China, **2** Key Laboratory of Intelligent Support Technology in Complex Environment, Ministry of Education, and School of Electronic and Information Engineering, Nanjing University of Information Science and Technology, Nanjing, China

* qingzeng2024@163.com

## Abstract

In the rapidly evolving landscape of the Internet of Things (IoT), traditional defense mechanisms struggle to counter sophisticated attribution attacks, especially under asymmetric information conditions. This paper introduces a novel Bayesian game framework—the Node-Based Attribution Attack-Defense Bayesian Game (NAADBG) Model—to address these challenges in IoT networks. The model incorporates a comprehensive set of attacker and defender profiles, capturing the complexities of real-world security scenarios. We develop a refined method for quantifying the payoffs of node-level attack-defense actions and explore the existence of a Mixed Strategy Bayesian Nash Equilibrium (MSBNE), enabling optimal defense strategy selection. Our simulations demonstrate that the NAADBG model significantly enhances network defense performance by optimizing resource allocation and preempting potential threats. This approach provides critical insights into developing proactive defense strategies against attribution attacks, contributing to more resilient IoT security frameworks. The results show that this method not only improves network defense performance but also presents practical applications in strengthening real-time IoT environments.

## Introduction

In today's increasingly interconnected world, the security of IoT networks has become a significant challenge, particularly at the node level, where individual devices are vulnerable to advanced and evolving cyber threats [1,2]. The complexity of protecting billions of IoT devices from attribution attacks—where attackers conceal their identity and origin—has exposed critical vulnerabilities in traditional defense mechanisms. This paper focuses on the novel challenge of defending IoT nodes from attribution attacks in an asymmetric information environment, where defenders have limited knowledge of attackers' strategies [3]. Addressing these vulnerabilities is crucial for maintaining the integrity of IoT networks, as compromised nodes can lead to severe consequences, including data theft, privacy violations, and large-scale network disruptions.

**Funding:** This work was supported by the National Key R&D Program of China (Grants No.

2021QY0700), the National Natural Science Foundation of China (Grants No. U21B2003, 62072250), Jiangsu Province Natural Science Foundation (Grants No. BK20230415), and Natural Science Foundation of the Jiangsu Higher Education Institutions of China (Grants No. 23KJB120007).

**Competing interests:** The authors have declared that no competing interests exist.

Traditional defense mechanisms, though effective in many areas, struggle to keep pace with the rapid development of attribution attack methods [4,5]. These methods, which often involve obfuscating attack sources and exploiting gaps in network defenses, present a serious challenge, especially in environments where defenders lack full knowledge of the attackers' strategies. Attackers can exploit this information asymmetry to their advantage, launching sophisticated attacks while remaining undetected. Therefore, it is imperative to develop defense mechanisms that can anticipate such threats, even with incomplete information.

Cyber adversaries frequently target IoT nodes, seeking to steal sensitive information such as financial data, personal records, or proprietary business information. In addition, attackers may compromise IoT components like edge nodes to orchestrate larger, more coordinated attacks. These threats highlight the necessity of defense strategies that go beyond reacting to attacks and proactively mitigate risks in environments where defenders are at an information disadvantage [6,7]. Furthermore, traditional defenses that rely on predefined rules or signature-based detection are often insufficient against zero-day attacks and advanced persistent threats. The ever-growing complexity of modern network environments makes it even more challenging to maintain up-to-date and comprehensive security measures.

To address these challenges, this paper introduces a Node-Based Attribution Attack-Defense Bayesian Game (NAADBG) model, which incorporates a game-theoretic approach to model the interactions between attackers and defenders in an asymmetric information setting. The NAADBG model enables defenders to anticipate attacker moves, assess the risks, and strategically allocate limited resources to enhance the defense of IoT nodes. The model not only considers the behaviors and strategies of both attackers and defenders but also accounts for the uncertainties present in real-world network scenarios. The main contributions are as follows:

- We propose a novel game-theoretic model tailored to IoT networks that addresses the problem of attribution attacks in asymmetric information environments. The model NAADBG incorporates both attacker and defender profiles to enhance its applicability to real-world security scenarios.
- We refine the quantification of payoffs for both attackers and defenders by assessing the impact of node-level attack-defense actions. We analyze the existence of a Mixed Strategy Bayesian Nash Equilibrium (MSBNE) and derive an optimal defense strategy selection method.
- Through simulation experiments, we evaluate the performance of the proposed model. The results demonstrate that the NAADBG model significantly improves network defense by optimizing resource allocation and preemptively mitigating threats.

The reminder of this paper is organized as follows. Sect 2 introduces the network attack-defense game model. Sect 3 specifies the payoff in attack-defense dynamics. Sect 4 derives the optimal defense strategy. Sect 4.3 gives the algorithm, the defender's strategy selection is determined by solving for the mixed strategy Bayesian Nash equilibrium. Simulations are conducted and then analysed in Sect 5. Finally, Sect 6 briefly concludes the paper.

## 1 Related work

While traditional mechanisms offer considerable protection, they often falter against novel and sophisticated cyberattacks that exploit previously unrecognized vulnerabilities. This limitation has driven the urgent need for defense mechanisms that are not only reactive but also predictive and proactive, capable of preempting threats before they manifest and optimizing defense strategies within the limited resources of network environments [8,9]. However,

traditional defense mechanisms are often limited by their reliance on predefined rules and signature-based detection, which can be insufficient against zero-day attacks and advanced persistent threats. Additionally, the increasing complexity and scale of modern network environments make it challenging to maintain up-to-date and comprehensive security measures. Moreover, existing literature often overlooks the perspective of attribution in cyberattacks, which can provide valuable insights into the strategic interactions between attackers and defenders. Attribution involves analyzing and tracking the source, path, and methods of cyberattacks to identify the attacker and their intentions. This process includes log analysis, traffic monitoring, and anomaly detection techniques. The goal of attribution is to find the attack source, understand the attacker's motives, and gather sufficient evidence for legal action or to develop more effective defense strategies. On the other hand, anti-attribution is a tactic used by attackers to conceal their true identity and attack path to avoid detection and attribution. Techniques for anti-attribution include using proxy servers, virtual private networks (VPNs), anonymous networks (such as Tor), and obfuscating and encrypting attack traffic. Attackers may also employ stepping-stone attacks, spoofing IP addresses, and utilizing botnets to further obscure their activities [10,11]. Therefore, understanding these strategic interactions between attackers and defenders is crucial for developing a comprehensive approach to network security.

Game theory plays an essential role in cybersecurity involving the conceptualization of conflicts and collaborations between attackers and defenders as strategic interactions [12,13]. Each participant aims to optimize their outcomes while minimizing risks and losses. Utilizing game-theoretic approaches, researchers and cybersecurity experts can forecast and scrutinize potential adversary actions, enabling the formulation of more robust defense strategies. The models typically abstract the nuanced dynamics of attacker-defender interactions into more manageable forms and often produce solutions that are probabilistic in nature. This abstraction and the inherent uncertainty of the results often hinder their direct application in real-world scenarios. The literature reveals several insights into the potential and challenges of employing game theory in network security [14–18].

Do et al. [14] review game-theoretic approaches for cybersecurity and privacy, categorizing their application into security and privacy aspects. The paper discusses the use of game theory in various scenarios such as cyber-physical security, communication security, and privacy, detailing the advantages and limitations from design to implementation of defense mechanisms. Liang and Xiao [15] discuss the application of cooperative and non-cooperative game models to handle network attacks, highlighting the relevance of such models in enhancing network defenses against sophisticated threats. Their review underscores the utility of game-theoretic approaches in understanding and mitigating complex network security challenges, which aligns with and could potentially enhance the Bayesian game framework proposed in this paper. Additionally, Manshaei et al. [16] provide a structured overview of game theory applications across various network security and privacy issues, suggesting the necessity for strategic decision-making to combat evolving cyber threats. Iqbal et al. [17] identify key challenges in game-theoretical modeling of network/cybersecurity, such as balancing complexity with practical applicability. These insights emphasize the growing importance of strategic frameworks like game theory in enhancing cyber defenses, enabling organizations to better prepare and respond to the dynamic nature of cyber threats and the strategic interactions that define modern cybersecurity landscapes. By leveraging game-theoretic principles, cyber defense strategies can evolve from reactive to proactive, anticipating attacker moves and optimizing defensive tactics to maintain security integrity and user privacy in an increasingly interconnected world. At the node level, by considering uncertainty, research in [19] strive

to develop optimal security strategies that mitigate the potential for unexpected network disruptions stemming from malicious attacks. Vijayalakshmi et al. [20] present a novel approach utilizing game theory to address the challenge of malicious packet dropping attacks in ad hoc networks. By designing an Intrusion Detection System (IDS) tailored to the unique characteristics of ad hoc networks, the proposed system effectively enhances network security by monitoring neighbor node behavior. A strategic defense resource allocation method is developed in [21] using an evolutionary game model to safeguard smart grids against potential cyber-attacks.

However, these discussions primarily focus on network defense strategies and attack models, without delving deeply into the implementation of attribution attack techniques. This limitation indicates a need for future research to pay more attention to effectively countering attackers' anti-attribution efforts in order to build a more comprehensive and robust node-level network security defense system. Moreover, these studies are conducted from the perspective of information symmetry, assuming that both attackers and defenders have equal access to information. This assumption overlooks the real-world scenario where information asymmetry often exists, putting defenders at a disadvantage. As a result, there is a significant gap in understanding how to develop effective defense mechanisms that account for this imbalance. The Bayesian game is a classic incomplete information game, where at least one participant's information is unknown to other participants, but the participants can have an initial judgment on the probability of other participants' types. This scenario aligns well with the realistic situation of network node attack-defense operations. Nevertheless, introducing Bayesian games with information asymmetry complicates the modeling process, making strategy optimization more challenging (Table 1).

## 2 NAADBG model design

In the realm of industrial IoT security, the interaction between attackers and defenders hinges on their mutual drive to maximize individual gains. Strategic decisions made by both parties are paramount, as they must weigh potential gains against the costs of their actions. Due to the network's complexity and the sensitive nature of its data, neither side can accurately ascertain the payoff of the opponent's strategy during confrontations. Nevertheless, by analyzing historical data, attackers and defenders can probabilistically infer their adversary's type. Employing the Harsanyi transformation, the uncertainty surrounding strategy payoffs can be translated into uncertainty about types, enabling each participant to be assigned a unique type reflecting their strategy and anticipated payoff. The ultimate aim for both parties

**Table 1. Comparison of related work in game-theoretic cybersecurity.**

| Author(s) | Focus | Contribution | Limitations |
|---|---|---|---|
| Do et al. (2017) [14] | Game theory in security and privacy | Review of cyber-physical and communication security | Limited discussion on attribution |
| Liang and Xiao (2013) [15] | Cooperative/non-cooperative games | Game theory in network defense | Assumes symmetric information |
| Manshaei et al. (2013) [16] | Game theory in network security | Strategic decision-making in cyber defense | Lacks focus on real-world asymmetry |
| Iqbal et al. (2019) [17] | Cybersecurity modeling challenges | Discusses complexity in game models | Limited on anti-attribution strategies |
| Zhang et al. (2022) [10] | Anti-attribution in IoT | Advanced anti-attribution techniques | No game-theory modeling |

is to safeguard their node network system, with the effectiveness of their attack or defense measured by the network system's value.

## 2.1 Basic assumptions

- **Rational Assumption:** Both attackers and defenders are considered rational entities that aim to maximize their payoff without engaging in actions that result in a net loss. Their payoffs are directly influenced by the strategies they adopt.

$$U_i(S_i, S_{-i}) \geq U_i(S_i', S_{-i}), \forall S_i' \neq S_i \tag{1}$$

Where $U_i$ is the utility function of player $i$, $S_i$ is the chosen strategy of player $i$, and $S_{-i}$ is the strategy of the opponent.

- **Type Assumption:** The strategy payoff's uncertainty is converted into type uncertainty using the Harsanyi transformation, facilitating a probabilistic analysis of opponent types.

$$P(T_i|S_{-i}), \forall T_i \in T \tag{2}$$

where $T_i$ is the type of player $i$, $P(T_i|S_{-i})$ is the probability distribution of types and the opponent's strategy $S_{-i}$.

- **Payoff Assumption:** The payoffs for both attackers and defenders are quantifiable through the security value of the network system, emphasizing the tangible impact of attack and defense strategies on the system's integrity and functionality.

$$U_i = f(\text{Security Value of Network}), \forall i \in \{A, D\} \tag{3}$$

where $U_i$ represents the payoff for player $i$, either attacker (A) or defender (D), quantified through the security value of the network.

## 2.2 Game model

The industrial internet's security dynamics, characterized by a continuous attacker-defender interaction, are modeled through an incomplete information stochastic game. This model simplifies to a two-player scenario, encapsulated in a 5-tuple representation NAADBG = $(N, T, S, P, U)$:

- **Players**:

$$N = \{N_A, N_D\} \tag{4}$$

$N_A$ represents the node attacker, and $N_D$ represents the network defender.

- **Types**:

$$T = \{T_A, T_D\} \tag{5}$$

where $T_A = \{T_1^A, T_2^A, \ldots, T_m^A\}$ for the attacker, e.g. advanced attacker, regular attacker, $T_D = \{T_1^D, T_2^D, \ldots, T_n^D\}$ for the defender, e.g., advanced defender, regular defender.

- **Strategies**:

$$S = \{S_A, S_D\} \tag{6}$$

$S_A = \{S_1^A, S_2^A, \ldots, S_k^A\}$ represents the set of attack strategies, e.g., phishing attack, DDoS attack, $S_D = \{S_1^D, S_2^D, \ldots, S_l^D\}$ represents the set of defense strategies, e.g., intrusion detection system, firewall rule update.

- **Beliefs**:

$$P = \{P_A, P_D\} \tag{7}$$

represents the prior beliefs of the attacker and defender about each other's types.
- **Payoff Functions**:

$$U = \{U_A, U_D\}$$

$U_A(T_j^A, S_k^A, S_g^D)$ and $U_D(T_i^D, S_k^A, S_g^D)$ represent the payoffs for the attacker and defender, respectively. Their own type are $T_j^A$ for attacker, $T_i^D$ for defender, and their chosen strategies are $S_k^A$ for the attacker, $S_g^D$ for the defender.

## 3 Payoffs in node-level attribution attack-defense

In node-level cyber conflict, the choices made by both attackers and defenders pivot on the calculus of payoffs. Thus, the effectiveness of any optimal cyber defense strategy hinges on the precise valuation of these payoffs, rooted in the tangible impacts of attack-defense strategies.

### 3.1 Foundations for payoff valuation

The essence of deriving an optimal defense strategy lies in accurately valuing the node-level attack-defense payoff. We introduce several definitions to describe the quantitative framework for this valuation.

**Definition 1.** Value of Network Systems. This denotes the worth of network resources, mirrored in the security attributes of network devices, namely confidentiality, integrity, and availability. The value of a network device $R$ is delineated as $R(C_1)$, $R(C_2)$, and $R(C_3)$, corresponding to these attributes respectively.

**Definition 2.** Impact of Attacks. This metric quantifies the effect of attacks on an IoT network system's value. The impact, denoted by $W$, is divided into $W(C_1)$, $W(C_2)$, and $W(C_3)$, reflecting the influence on the confidentiality, integrity, and availability of network devices.

**Definition 3.** Probability of Attack Success. Represented by $\theta$, this metric captures the likelihood of an attack breaching defenses to exploit information resources. It considers the detection probability $\lambda$ and the defense success rate $\beta$, with the failure of an attack contingent upon both detection and successful defense, leading to $\theta = 1 - \lambda\beta$.

**Definition 4.** Attack Payoff. This represents the gains an attacker secures from a successful assault. An unsuccessful attack yields defense insights, albeit at the cost of leaving traces within the defense system. Such traces prompt the defender to adjust defenses, concentrating on exposed vulnerabilities. Consequently, attackers reap benefits solely from successful attacks, with payoffs calculated based on inflicted damage to network value.

**Definition 5.** Defense Payoff. This quantifies the gains from defensive actions, aimed at safeguarding network system value. The payoff is ascertainable regardless of the defense action's outcome.

- Successful defense actions directly protect network value, yielding immediate payoffs based on the preserved system value.
- Unsuccessful defenses, while not directly safeguarding system value, enable defenders to gather attack intelligence, potentially enhancing future defense success rates and generating indirect benefits. Such indirect payoffs derive from both the defense's improved efficacy and the value of potentially safeguarded network systems.

## 3.2 Valuation of model payoffs

The attacker node type $T_j^A$ and strategy $A_h(S_k^A)$ target the defense. Conversely, the defender node type $T_i^D$ and strategy $D_q(S_g^D)$ aim to protect the network.

The expected value for the attacker in security attribute $C_x$, given a successful attack, is expressed as:

$$E_{A_{h1}}(C_x) = (1 - \lambda_h \beta_q) W_{A_h}(C_x) R(C_x) \qquad (8)$$

where the detection probability by the defender is $\lambda_h$, and $\beta_q$ signifies the defense success rate.

If the attack fails, the expected value $E_{A_{h2}}(C_x)$ of the attacker can be given as

$$E_{A_{h2}}(C_x) = \mu_h \lambda_h \beta_q W_{A_h}(C_x) R(C_x) \qquad (9)$$

Accordingly, the attacker's payoff, upon executing action $A_h(S_k^A)$, is formulated as:

$$U_{A_h} = \sum_{x=1}^{3} (1 - \lambda_h \beta_h + \mu_h \lambda_h \beta_q) W(C_x) R(C_x) - B(A_h) \qquad (10)$$

Here, $W(C_x)$ denotes the attack node's impact on network device security attributes $C_x$, $R(C_x)$ represents the value of the targeted device, and $B(A_h)$ indicates the cost of the attack action $A_h(S_k^A)$. This includes the expenses related to developing and deploying attack tools, the time and effort spent, and the resources consumed in bypassing defense mechanisms. For instance, in the case of a Distributed Denial of Service (DDoS) attack, $B(A_h)$ may encompass the cost of purchasing or renting botnet devices, the fees for controlling these devices, and the bandwidth expenses incurred during the attack.

For the defender node, the expected value following a successful defense action in security attribute $x$ is given by:

$$E_{D_q}(C_y) = \lambda_h \beta_q W(C_x) R(C_x) \qquad (11)$$

The defender node's payoff from action $D_q(S_g^D)$ is determined as:

$$U_{D_q} = \sum_{y=1}^{3} \lambda_h \beta_q + W(C_y) R(C_y) - B(D_q) \qquad (12)$$

where $\mu_q$ is the discount factor for a failed defense and $B(D_q)$ denotes the defense action cost. This includes the expenses for implementing and maintaining defense mechanisms, the time and effort required for continuous monitoring, and the resources allocated for updating and patching systems. Deploying an Intrusion Detection System (IDS) might include costs related to purchasing the IDS software and hardware, ongoing maintenance and updates, and the manpower needed for monitoring and responding to detected threats.

When diverse defense actions impact an attack strategy, the attacker node's payoff is the lowest of the payoffs, while the defender node's is the highest. The strategic confrontation payoffs, when both parties opt for strategies $(S_k^A, S_g^D)$, are articulated as:

$$U_A\left(T_j^A, S_k^A, S_g^D\right) = \sum_{h=1}^{l} P_k^A(A_h) \sum_{q=1}^{\gamma} P_g^D(D_q) U_{A_h}$$

$$= \sum_{h=1}^{l} P_k^A(A_h) \sum_{q=1}^{\gamma} P_g^D(D_q) \left( \sum_{x=1}^{3} \left( 1 = \lambda_h \beta_q \right. \right.$$

$$+\mu_h \lambda_h \beta_q \big) W\left(C_x\right) R\left(C_x\right) - B\left(A_h\right)\big)$$

$$U_D\left(T_i^D, S_k^A, S_g^D\right) = \sum_{q=1}^{r} P_g^D\left(D_q\right) \sum_{h=1}^{l} P_k^A\left(A_h\right) U_{D_q} \qquad (13)$$

$$= \sum_{q=1}^{r} P_g^D\left(D_q\right) \sum_{h=1}^{l} P_k^A\left(A_h\right) \left(\sum_{y=1}^{3} \lambda_h \beta_q W\left(C_y\right) R\left(C_y\right)\right.$$

$$\left. - B\left(D_q\right)\right)$$

where in this framework, $P_k^A\left(A_h\right)$ denotes the probability that the attacker node will select action $A_h\left(S_k^A\right)$ when adopting attack strategy $S_k^A$. Similarly, $P_g^D\left(D_q\right)$ represents the probability that the defender node will opt for defense action under the chosen defense strategy $S_g^D$.

## 4 Optimal defense strategy selection

To ensure the methodology aligns with the proposed optimal defense strategy algorithm based on the attack-defense game, we have validated the algorithm's core structure and performance. The NAADBG model integrates both attack and defense actions into a Bayesian game framework, enabling the optimal strategy selection for defenders facing node-level attacks in IoT environments. The algorithm computes the mixed strategy Bayesian Nash equilibrium and ranks defense strategies based on their effectiveness in response to varying attacker profiles. Below, we outline the specific aspects of the validation.

### 4.1 Nash equilibrium analysis

Within the Node-Level Attack-Defense Game (NAADBG), both the attacker and defender node aim to maximize their respective payoffs, which are influenced by their types and prior beliefs. Achieving a balance, or Nash equilibrium, under pure strategy is not always possible, thus mixed strategies often serve as a practical approach to analyze equilibrium scenarios.

**Definition 6. Mixed strategy.** The attacker selects a pure attack strategy with the probability $f_k^A\left(T_j^A\right)$. Upon selecting the attack strategy, the constraints

$$0 \le f_k^A\left(T_j^A\right) \le 1, \quad \sum_{k=1}^{k_1} f_k^A\left(T_j^A\right) = 1$$

need to be considered. $F_A\left(T_j^A\right) = \{f_1^A\left(T_j^A\right), f_2^A\left(T_j^A\right), \ldots, f_{k_1}^A\left(T_j^A\right)\}$ is a mixed strategy of the attacker node under the type $T_j^A$. Similarly, $F_D\left(T_i^D\right) = \{f_1^D\left(T_i^D\right), f_2^D\left(T_i^D\right), \ldots, f_{k_2}^D\left(T_i^D\right)\}$ is also a mixed strategy of the defender node under type $T_i^D$.

**Definition 7. Mixed strategy Bayesian Nash equilibrium.** $F_A^*\left(T_j^A\right) = \{f_1^A\left(T_j^A\right), f_2^A\left(T_j^A\right), \ldots, f_{k_1}^A\left(T_j^A\right)\}$ denotes the mixed strategy of the attacker node, and $F_D\left(T_i^D\right) = \{f_1^D\left(T_i^D\right), f_2^D\left(T_i^D\right), \ldots, f_{k_2}^D\left(T_i^D\right)\}$ is the mixed strategy of the defender node. If the mixed strategy $\left(F_A^*\left(T_j^A\right), F_D^*\left(T_i^D\right)\right)$ meets the constraints, i.e.,

$$\left(\sum_{i=1}^{m} P_A\left(T_i^D \big| T_j^A\right) U_A\left(T_j^A, \ F_A^*\left(T_j^A\right), F_D^*\left(T_i^D\right)\right)\right)$$

$$\ge \sum_{i=1}^{m} P_A\left(T_i^D \big| T_j^A\right) U_A\left(T_j^A, F_A^*\left(T_j^A\right), F_D^*\left(T_i^D\right)\right) \qquad (14)$$

and

$$\left( \sum_{j=1}^{n} P_D \left( T_j^A \middle| T_i^D \right) U_D \left( T_i^D, \ F_A^* \left( T_j^A \right), F_D^* \left( T_i^D \right) \right) \right)$$

$$\geq \sum_{i=j}^{m} P_D \left( T_j^A \middle| T_i^D \right) U_D \left( T_i^D, F_A^* \left( T_j^A \right), F_D^* \left( T_i^D \right) \right)$$

(15)

then the Bayesian Nash equilibrium can be achieved.

In the game, the mixed strategy $\left( F_A^* \left( T_j^A \right) \right), F_D^* \left( T_i^D \right) \right)$ is adopted when both parties achieve a state of equilibrium. The method for calculating the mixed strategy Bayesian Nash equilibrium can be mathematically delineated as follows:

$$\arg \max f \left( S_A, S_D, U_A, U_D, T_A, T_D \right)$$

$$= \sum_{i=1}^{m} P_A \left( T_i^D \middle| T_j^A \right) U_A \left( T_j^A, F_A^* \left( T_j^A \right), F_D^* \left( T_i^D \right) \right)$$

$$+ \sum_{j=1}^{m} P_D \left( T_j^A \middle| T_i^D \right) U_D \left( T_i^D, F_A^* \left( T_j^A \right), F_D^* \left( T_i^D \right) \right) - v_1 - v_2;$$

$$\sum_{i=1}^{m} P_A \left( T_i^D \middle| T_j^A \right) U_A \left( T_j^A, F_A \left( T_j^A \right), F_D^* \left( T_i^D \right) \right) \leq v_1;$$

$$\sum_{j=1}^{n} P_D \left( T_j^A \middle| T_i^D \right) U_D \left( T_i^D, F_A^* \left( T_j^A \right), F_D \left( T_i^D \right) \right) \leq v_2;$$

$$\sum_{i=1}^{m} P_A \left( T_i^D \middle| T_j^A \right) \in [0,1], \sum_{j=1}^{n} P_D \left( T_j^A \middle| T_i^D \right) \in [0,1];$$

$$\sum_{i=1}^{m} P_A \left( T_i^D \middle| T_j^A \right) = 1, \sum_{j=1}^{n} P_D \left( T_j^A \middle| T_i^D \right) = 1.$$

(16)

**Theorem 1:** The mixed strategy Bayesian Nash equilibrium of the NAADBG exists.

*Proof:* First, the NAADBG consists of several independent and similar Bayesian games. Since each independent Bayesian game is a finite game, the basic theorem of Bayesian games [22] indicates that a mixed strategy Nash equilibrium exists. Furthermore, according to the definition of the network attack-defense game, its payoff function is a convex function based on transition probabilities and payoff functions. According to the existence theorem of equilibrium strategies in finite stochastic games, we can prove that the mixed strategy Bayesian Nash equilibrium of NAADBG exists.

Chatterjee B [22], Cheng L et al. [23], Wang et al. [12] have reported that the solution to the mixed strategy Bayesian Nash equilibrium can be formulated as a standard nonlinear programming problem. According to the game theory, the mixed strategy $\left( F_A^* \left( T_j^A \right) \right), F_D^* \left( T_i^D \right) \right)$ can be selected when both sides reach equilibrium state. Therefore, the equilibrium strategy solution for NAADBG can be equivalently transformed into the problem of finding the optimal value of a nonlinear programming problem. By forming a quadratic programming problem that combines the objective functions, we satisfy the constraints of the objective functions. ∎

## 4.2 Optimal defense strategy selection method

Selecting the optimal defense strategy in a network security game is a complex process, especially when dealing with incomplete information. In realistic node security defense scenarios,

defender nodes must choose strategies based on limited resources and incomplete knowledge about attacker nodes' strategies. Traditional game-theoretic methods typically yield mixed strategies as optimal solutions. However, for practical applications where network managers may prefer to implement pure strategies, a new approach is necessary. We introduce the concept of defense effectiveness as a criterion that balances the payoff of a defense strategy against an attack strategy when both parties have reached a Nash Equilibrium.

To measure the effectiveness of a defense in each security state, we define a utility function based on the payoff matrix and prior probabilities of defender strategies. In the context of node-level attack-defense games, we consider $F_D^*(T_i^D)$ to be the probability that a defender of type $T_{D_i}$ selects a defense strategy, and $P_D(T_j^A \mid T_i^D)$ is the prior belief of the defender about the attacker type. The payoff for a chosen defense strategy is $U_D(T_i^D, S_k^A(T_j^A), S_g^D)$.

The defense effectiveness of strategy $S_g^D$ can be quantified as follows:

$$E(S_g^D) = \sum_{j=1}^{n} P_D(T_j^A \mid T_i^D) \sum_{k=1}^{k_2} U_D(T_i^D, S_k^A(T_j^A), S_g^D) F_D^*(T_i^D) \qquad (17)$$

Upon calculating the effectiveness of all defense strategies, the effect of these strategies against the possible attack actions at equilibrium can be ascertained through defense effectiveness. The defense strategies can then be ranked, and the most effective strategy can be selected given the network resources at hand.

## 4.3 Algorithm description

**Algorithm 1** Optimal defense strategy selection algorithm based on attack defense game.

**Require:** Network attack-defense game model NAADBG
**Ensure:** Optimal defense strategy
1: Initialize the model parameter NAADBG = $\{\mathcal{N}, \mathcal{T}, \mathcal{S}, P, U\}$
2: Construct attack and defense type set $\mathcal{T}_A$ and $\mathcal{T}_D$
3: Construct attack and defense strategy set $\mathcal{S}_A$ and $\mathcal{S}_D$
4: Obtain attack and defense prior belief set $P_A$ and $P_D$
5: **while** $S_k^A \in \mathcal{S}_A, S_j^D \in \mathcal{S}_D$ **do**
6:     Calculate the payoffs of attack-defense strategies
7:     Calculate the payoff of attack strategy $U_A(T_j^A, S_k^A, S_j^D)$;
8:     Calculate the payoff of defense strategy $U_D(T_i^D, S_k^A, S_j^D)$;
9: **end while**
10: The mixed strategy $(F_A^*(T_j^A), F_D^*(T_i^D))$ can be obtained;
11: Return $\arg\max(E(S_g^D))$; //output the optimal defense strategies of each security state

Based on the quantification of attack-defense payoffs, the defender's strategy selection is determined by solving for the mixed strategy Bayesian Nash equilibrium. The defense effectiveness is quantified considering the defender's prior beliefs and the payoff matrix. The optimal defense strategy is then selected based on the criterion of maximum defense effectiveness, differentiating this approach from classical algorithms that rely on mixed strategies. Our model advocates for the selection of a pure strategy, which enhances practical operability. By selecting the optimal strategy beforehand, the network is proactively defended against potential security threats. The key steps of this novel defense strategy include: (1) quantifying payoffs for each potential defense strategy; (2) solving for the Bayesian Nash equilibrium to

ascertain feasible strategies; (3) evaluating the effectiveness of each strategy based on the equilibrium analysis; and (4) implementing the defense strategy with the highest effectiveness to activate the network's defenses prior to the occurrence of security threats.

The time complexity of the proposed algorithm primarily focuses on two aspects: game payoff quantification and the Nash equilibrium solution process. During the game payoff quantification, if the attacker's type is $T_j^A$, then the attack strategy $S_k^A$ is selected to target. It is well known that this process has a certain complexity, which follows the order of $O(|T_D| \cdot |S_D(t_D)|)$. The number of attacker types is $|T_A|$, and the number of attack strategies for each type is $|S_A(t_A)|$. Therefore, the time complexity of the game payoff quantification can be expressed as $O(|T_A| \cdot |T_D| \cdot |S_A(t_A)| \cdot |S_D(t_D)|)$. In the Nash equilibrium solution process, the game model used by this algorithm is a non-zero-sum and static game. It has been demonstrated that finding a Nash equilibrium is a PPAD-incomplete problem. In a practical network environment, it is unnecessary to find all Nash equilibrium solutions. The algorithm can be terminated to enhance efficiency once an appropriate Nash equilibrium is found. In a realistic network attack-defense scenario, both the types of attackers and defenders are small constants. Thus, the complexity of the proposed algorithm can satisfy the requirements of network attack and defense.

Note that in scenarios where attackers fail to achieve their optimal outcomes, particularly under asymmetric information, the NAADBG model dynamically adjusts the defender's strategies by leveraging Bayesian updating. As the attacker deviates from optimal behavior, the defender revises their prior beliefs about the attacker's strategy, enabling more accurate predictions of future actions. This adaptability ensures that even in cases of suboptimal attacker actions, the model remains effective at selecting defense strategies that maximize network protection.

## 5 Simulation experiment and analysis

To assess the effectiveness of the proposed method for selecting optimal defense strategies against node-level attacks, an experimental network system was configured following the standard experimental network environments. The network's structure separates an internal network, safeguarded by a firewall, from potential external threats. The firewall configuration restricts external hosts to interacting solely with designated mail and web servers. Within the internal network, specific servers and administrators maintain the capability to access and manage the database server, ensuring operational control.

Reflecting on historical attack patterns, we categorize attackers into two distinct profiles: attribution depth and attribution efficiency.

- **Attribution Depth**: This attacker profile is defined by a cautious approach that aims to deeply infiltrate systems at the node level while remaining undetected. The focus is on stealth and sustained access rather than quick results, with efforts concentrated on erasing any traces that could lead back to them. Specific measures can be: 1) Advanced Persistent Threats (APTs), i.e., using sophisticated malware that remains hidden for long periods. 2) Data Exfiltration Techniques, i.e., tealthily transferring data in small amounts to avoid triggering alarms.
- **Attribution Efficiency**: This profile describes an attacker who aims for quick and impactful results using minimal resources. The focus is on achieving goals swiftly and possibly openly, with less concern for hiding their tracks or establishing long-term access. Specific measures include: 1) DDoS Attacks: Quickly overwhelming network nodes to disrupt service.

2) Exploiting Zero-Day Vulnerabilities: Using newly discovered vulnerabilities to launch attacks before patches are available.

Defenders can be classified into two distinct groups based on the effectiveness and expenditure of their defense strategies: namely, "senior defenders" and "primary defenders." Senior defenders are identified by their readiness to engage in more expensive defense measures to attain a robust level of protection at critical nodes. In contrast, a primary defender tends to implement more cost-effective defense tactics, focusing on achieving their defensive objectives within their budgetary constraints. For instance, Redundancy and Failover Systems to implement redundant systems to ensure continuity in case of node failure. Defenders typically choose a combination of various defensive measures, with different types of defenders selecting different actions. To determine the appropriate defensive actions from the defensive behavior library, factors such as costs, impacts, and expert advice are considered. The simulation environment replicates a standard network system with predefined configurations for internal and external network structures, focusing on node-level security. We incorporate historical attack patterns targeting individual nodes to realistically categorize attacker profiles and accordingly define defender categories as 'senior defenders' and 'primary defenders'. The model parameters were initialized following the constructs of the NAADBG (Table 2).

The choice of network attack and defense strategies can be defined as follows: traditional attacks, denoted as $M_1$, and deep infiltration attacks, represented by $M_2$. The probability distribution for an attacker launching a traditional attack is $p$, while that for performing a deep infiltration attack is $1-p$. Regarding strategy selection, the options include launching an attack or not launching an attack. The probability of not launching an attack is defined as $\omega$, and the probability of launching an attack is $1 - \omega$.

On the defense side, we have primary defenders, denoted as $N_1$, and senior defenders, represented by $N_2$. The probability distribution for a primary defender responding to an attack is $q$, while for a senior defender, it is $1-q$. The strategies include defending or not defending. The probability of not defending is denoted as $\varpi$, while the probability of defending is $1 - \varpi$.

Since the players know the opponent's strategy, Bayesian rules are applied to obtain the payoffs of the players in the game, allowing us to calculate the expected maximum payoffs for all participants. Therefore, the set of attack strategies can be divided into four cases: $\{(A_1, A_1), (A_1, A_2), (A_2, A_1), (A_2, A_2)\}$. Here, $(A_1, A_1)$ represents the strategy where the attacker does not launch an attack regardless of the type of attack. $(A_1, A_2)$ indicates that the attacker does not launch a traditional attack but launches a deep infiltration attack. $(A_2, A_1)$ is the opposite of $(A_1, A_2)$, and $(A_2, A_2)$ is the opposite of $(A_1, A_1)$.

Similarly, the set of defender strategies can be divided into four cases: $\{(B_1, B_1), (B_1, B_2), (B_2, B_1), (B_2, B_2)\}$. $(B_1, B_1)$ represents the strategy where the defender does not take any defensive action regardless of the defender type. $(B_1, B_2)$ indicates that the primary defender does not defend, but the senior defender does. $(B_2, B_1)$ is the opposite of $(B_1, B_2)$, and $(B_2, B_2)$ is the opposite of $(B_1, B_1)$.

**Table 2. Descriptions of formula variables in the NAADBG model.**

| Symbols | Descriptions |
| --- | --- |
| $M_1(M_2)$ | Attribution depth or attribution efficiency attack |
| $N_1(N_2)$ | Senior defender or primary defender |
| $A_2(A_1)$ | Attacker launches attack (or not) |
| $B_2(B_1)$ | Defender defenses (or not) |

Fig 1 demonstrates the effect of $N$ (the number of defense resources) on the payoffs of defenders and attackers. As $N$ increases, the payoffs for defenders, particularly senior defenders (SD), exhibit a steady rise. This indicates that increasing the quantity of defense resources significantly enhances defense capabilities. In a real-world network context, this suggests that deploying more resources, such as additional firewalls or intrusion detection systems, can noticeably strengthen network defenses.

Conversely, the payoffs for attackers, especially Attribution Depth Attackers (ADA), decrease as $N$ increases. This suggests that as the defense capabilities grow stronger with more resources, the attackers' potential gains diminish, particularly for sophisticated, stealthy attacks. This demonstrates the effectiveness of scaling defense resources to mitigate advanced cyber threats.

Fig 2 illustrates the effect of the defense success rate, $\beta$, on the payoffs in the Node-Level Attribution Attack-Defense Bayesian Game (NAADBG). As $\beta$ increases, senior defenders (SD) who do not employ active defense (ND) maintain high payoffs, reflecting the robustness of their defensive measures. In contrast, primary defenders (PD) with active defense (D) experience a moderate decrease in payoffs, suggesting diminishing returns on lower-level defense actions.

For attackers, the payoffs, particularly for Attribution Depth Attackers (ADA) without active attacks (NA), show a sharp decline as $\beta$ increases, dropping from near zero to significantly negative values. This result shows that as the defense success rate improves, attackers face increasingly severe losses. The trend supports the model's predictive capability: higher defense efficiency drastically reduces the potential benefits for attackers, even when they attempt to evade detection.

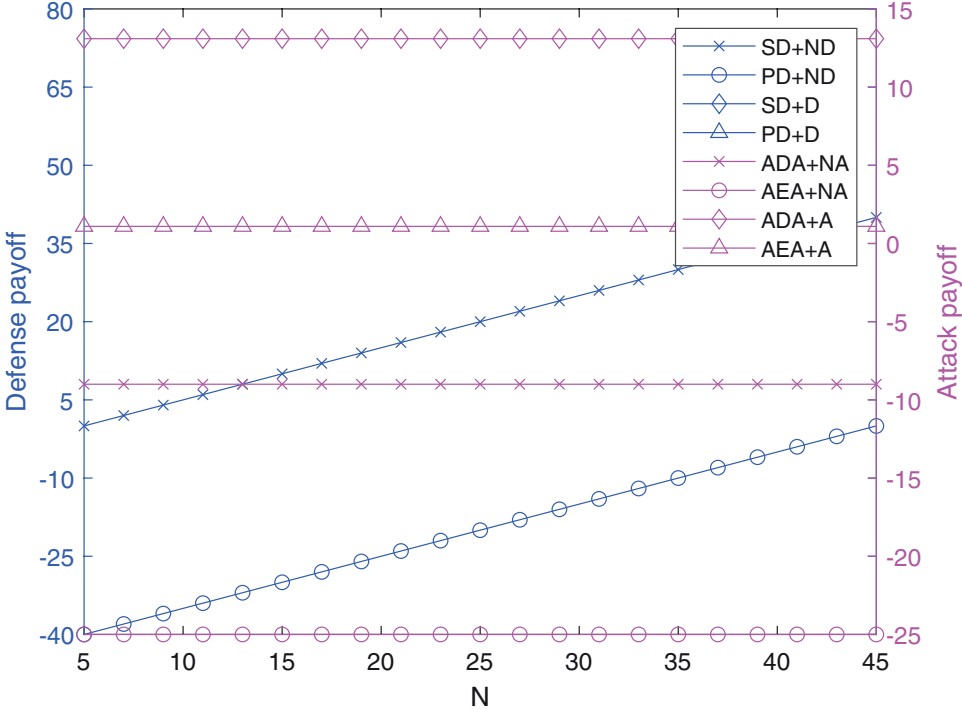

**Fig 1. The impact of $N$ on payoffs in NAADBG.**

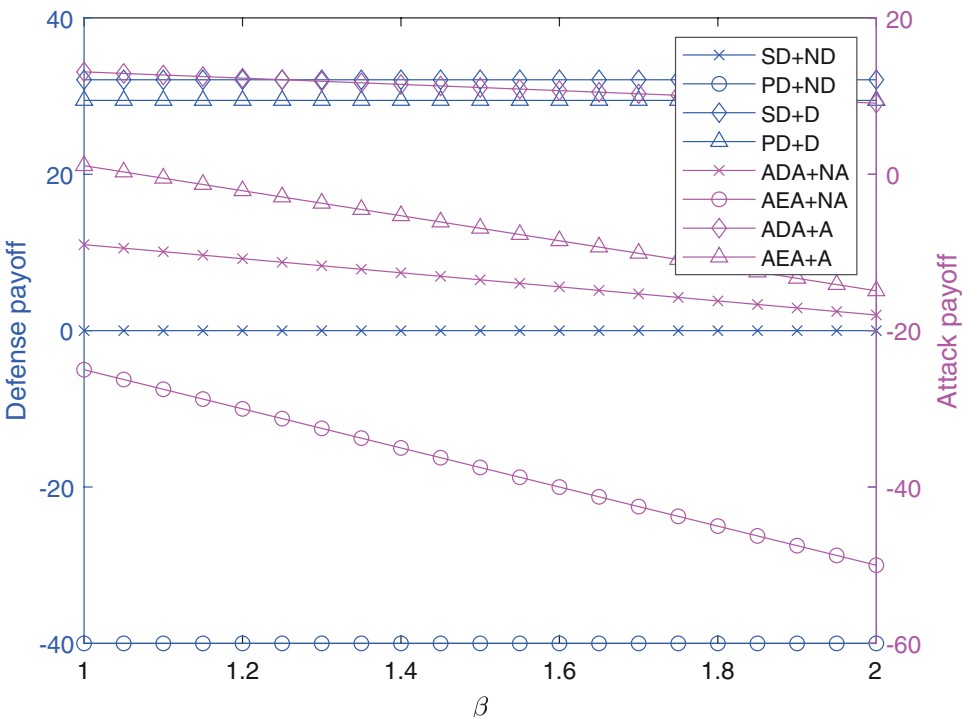

**Fig 2. The impact of $\beta$ on payoffs in NAADBG.**

Fig 3 explores how varying the detection probability, $\lambda$, affects the payoffs in NAADBG. Senior defenders (SD) without active defense (ND) retain a high, stable payoff, highlighting the importance of detection in reinforcing strong defenses. Primary defenders (PD) with active defense (D) see relatively stable payoffs around 200 units.

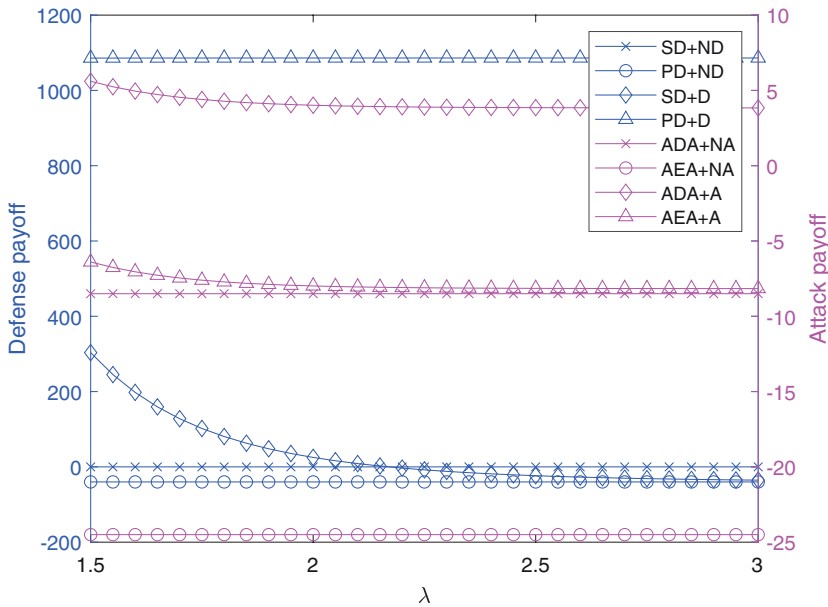

**Fig 3. The impact of $\lambda$ on payoffs in NAADBG.**

On the other hand, attackers experience significant declines in payoffs as $\lambda$ increases. ADA without active attacks (NA) sees a drastic drop from zero to negative values, while other strategies also exhibit declining returns. This underscores the critical role of detection probability in cyber defense: as detection capabilities improve, attackers are forced into negative payoffs, making it more difficult to sustain successful attacks.

Fig 4 illustrates that increasing the network system's value, $R(C_x)$, impacts strategic payoffs in the NAADBG model. Senior defenders (SD) without defense (ND) shows a slight decrease in payoff, while primary defenders (PD) with defense (D) experience a more pronounced decline. Attackers, both ADA without active attack (NA) and AEA, regardless of attack (A), also face reduced payoffs, with ADA remaining stable and AEA showing a steep decline. Average payoff differences highlight that ADA+A and AEA+A strategies have a consistent difference of 10, AEA+NA has 11.27, PD+ND shows 14.09. These results indicate that higher $R(C_x)$ reduces defense payoffs, likely due to higher protection costs, while significantly diminishing attack payoffs, reflecting increased difficulty in successful exploitation.

Fig 5 explores how the cost of attack strategies, $B(A_h)$, influences payoffs. As $B(A_h)$ increases, primary defenders (PD) with defense (D) see a decrease in payoffs, dropping from 1100 to 800, while PD without defense (ND) remains relatively stable around zero.

Interestingly, the payoffs for attackers remain stable as $B(A_h)$ increases, especially for ADA and AEA strategies without active attacks (NA), maintaining consistent values. This suggests that while the cost of executing attacks increases, it does not significantly impact the attackers' immediate payoffs, highlighting the importance of balancing defense resource allocation against different attack costs.

Fig 6 examines how the attack impact, $W(C_x)$, affects the payoffs in NAADBG. As $W(C_x)$ increases, the payoffs for senior defenders (SD) rise steadily from negative values to over

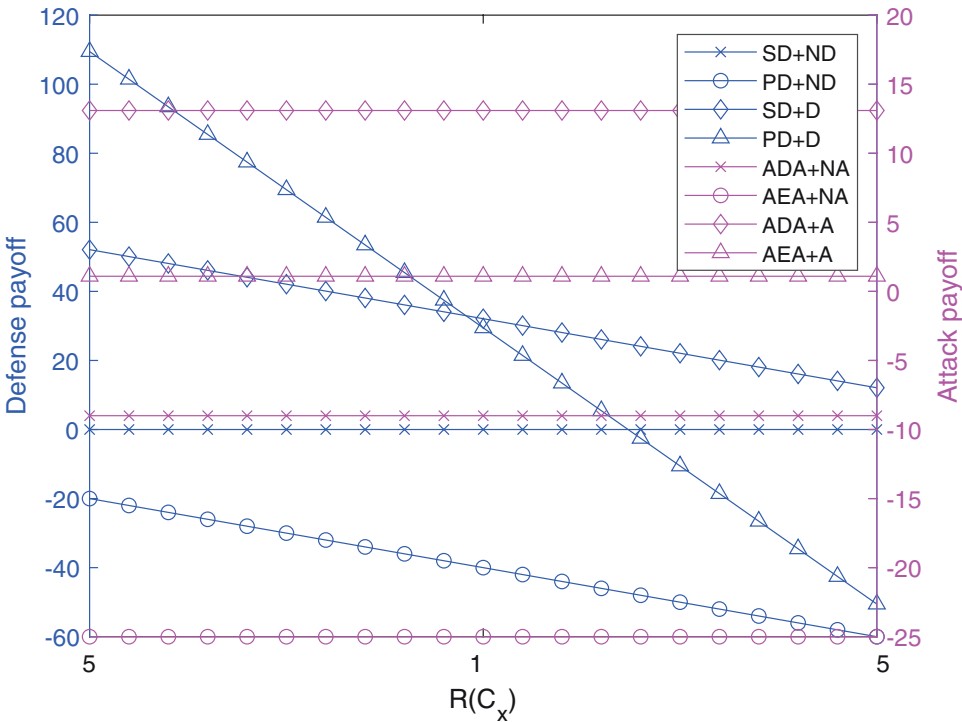

**Fig 4. The impact of $R(C_x)$ on payoffs in NAADBG.**

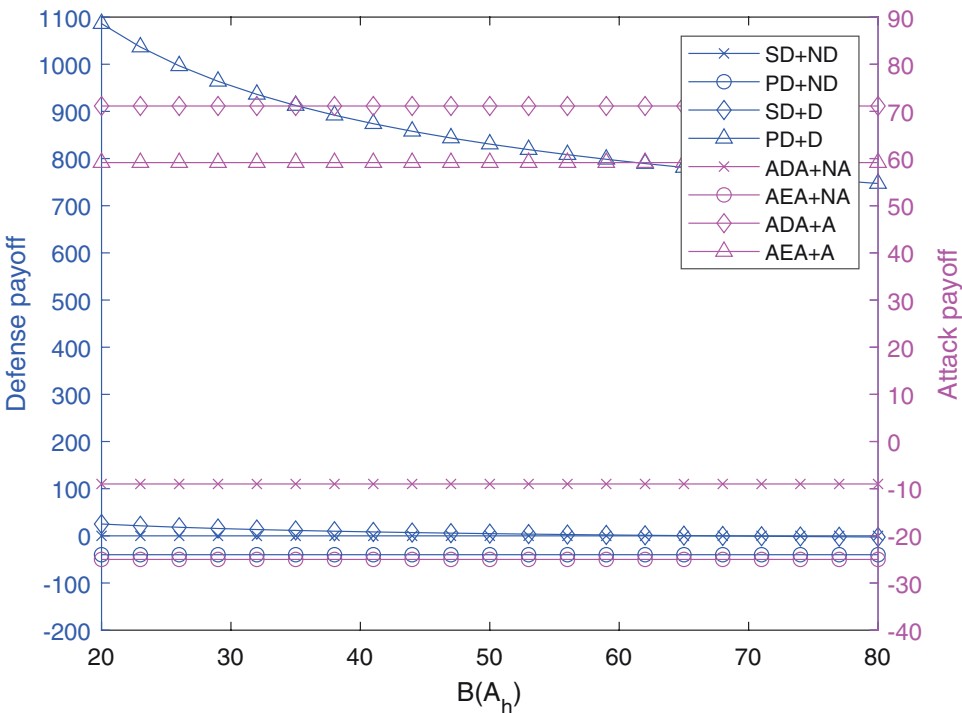

**Fig 5. The impact of $B(A_h)$ on payoffs in NAADBG.**

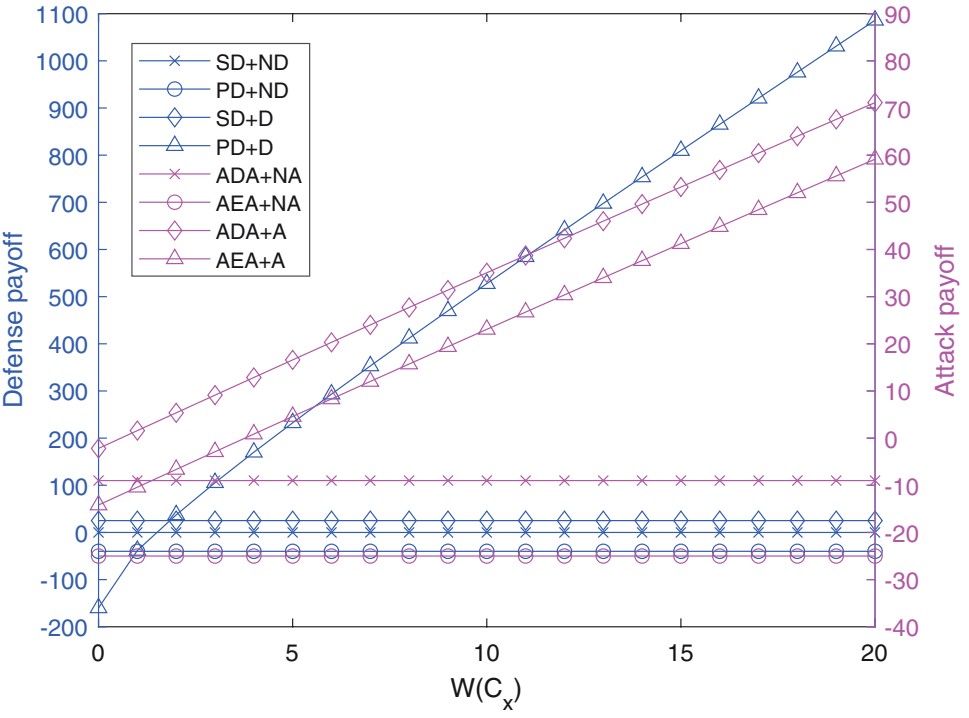

**Fig 6. The impact of $W(C_x)$ on payoffs in NAADBG.**

1100, indicating that defense becomes more effective as the potential attack impact increases. Senior defenders who do not employ active defense (ND) maintain consistent payoffs, suggesting that sophisticated defense strategies can still mitigate attack impacts without active intervention.

On the other hand, both ADA and AEA strategies see rising payoffs as $W(C_x)$ grows, reflecting the increasing potential rewards for successful attacks. This highlights the need for defenders to adjust their strategies based on the potential impact of attacks, ensuring that high-value targets receive more robust protection to counter attackers' increased motivations.

We compare with three baseline methods:

- Traditional Game Theory [14]: Assumes complete information and static strategies, limiting its adaptability to advanced threats.
- Cooperative Game Theory [15]: Focuses on resource sharing among defenders, which enhances defense but relies on cooperation.
- Hybrid Game Model [20]: Combines cooperative and non-cooperative approaches, offering flexibility but struggling with complex, asymmetric information scenarios.

Based on the baseline comparison, as shown in Fig 7, it is evident that the NAADBG model consistently outperforms other methods, including Traditional Game Theory, Cooperative Game Theory, and the Hybrid Game Model. The NAADBG model demonstrates a sharp increase in defense payoffs as the defense success rate ($\beta$) rises, indicating its superior

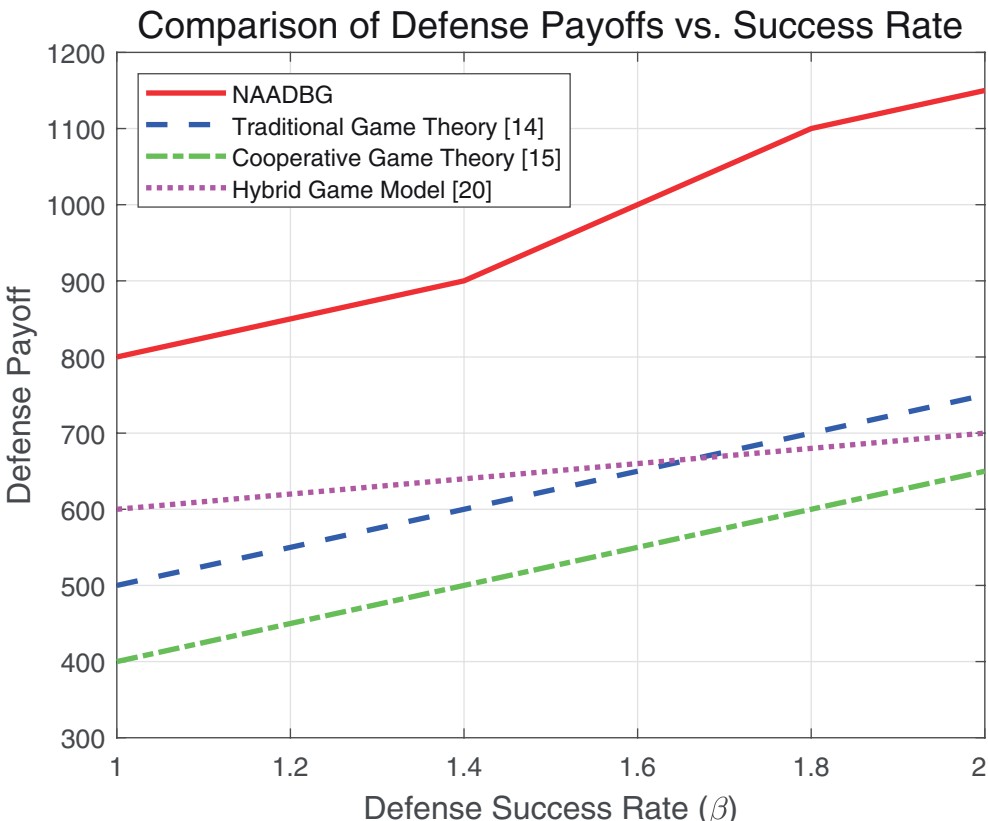

**Fig 7. Comparison of defense payoff vs. success rate.**

efficiency in resource allocation and adaptability to evolving threats in IoT environments. In contrast, other models show a more gradual and linear increase in payoffs, suggesting that they are less effective at handling complex node-level attribution attacks. The NAADBG's ability to anticipate and counteract sophisticated attacks with limited information makes it the most robust model for optimizing defense strategies.

## 6 Conclusion

This paper introduces a Node-based Attribution Attack-Defense Bayesian Game (NAADBG) model to address attribution attacks in IoT environments under information asymmetry. By incorporating diverse attacker and defender profiles, the model closely aligns with real-world network security challenges. We enhance payoff quantification, analyze the existence of mixed strategy Bayesian Nash equilibrium, and derive an optimal defense strategy selection method. Simulation results demonstrate the model's effectiveness in improving defense performance, particularly in resource-constrained environments. However, integrating the NAADBG model into existing security infrastructures may pose challenges, such as computational overhead and compatibility with current systems, which require further research for practical deployment.

## Author contributions

**Data curation:** Xin Sun.

**Methodology:** Xin Sun.

**Resources:** Guangjie Liu.

**Software:** Jun Chen, Guangjie Liu.

**Validation:** Wen Tian, Guangjie Liu.

**Visualization:** Wen Tian.

**Writing – original draft:** Jun Chen.

**Writing – review & editing:** Jun Chen.

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
