## [Decision Letter · Decision Letter 0]

26 Aug 2024

PONE-D-24-27183A Node-Based Defense Strategy against Attribution Attack in IoT With Asymmetric InformationPLOS ONE

Dear Dr. Chen,

Thank you for submitting your manuscript to PLOS ONE. After careful consideration, we feel that it has merit but does not fully meet PLOS ONE’s publication criteria as it currently stands. Therefore, we invite you to submit a revised version of the manuscript that addresses the points raised during the review process.

We look forward to receiving your revised manuscript.

Kind regards,

Brij Bhooshan Gupta

Academic Editor

PLOS ONE

3. In the online submission form, you indicated that [Data are available when requested.].

Reviewers' comments:

Reviewer's Responses to Questions

**Comments to the Author**

1. Is the manuscript technically sound, and do the data support the conclusions?

Reviewer #1: Yes

Reviewer #2: Yes

2. Has the statistical analysis been performed appropriately and rigorously? 

Reviewer #1: Yes

Reviewer #2: Yes

3. Have the authors made all data underlying the findings in their manuscript fully available?

Reviewer #1: No

Reviewer #2: Yes

4. Is the manuscript presented in an intelligible fashion and written in standard English?

Reviewer #1: Yes

Reviewer #2: Yes

5. Review Comments to the Author

Reviewer #1: The Author has made an effort to study A Node-Based Defense Strategy against Attribution Attack in IoT With Asymmetric Information. The paper is quite important and relevant , but I found the technical contribution of this paper is need to be relook. Also, some of the following issues need to be addressed for the improvement of the paper.

Title is needed to relook and make it more appropriate in the view of contribution. Further, The paper's abstract should focus on the importance of addressing this issue. The highlights of the results and how the reader might benefit from the paper's material should also be included.

The Introduction section needs to relook to improve its quality and readability. Further¸ it is suggested to add an RELATED STUDY Section and need to add the existing work and make an comparative table to address the various key gaps in existing work which are going to address through this paper.

Performance reviews are mostly used to summarize the findings of interpretation and inference, highlighting new discoveries and innovations. So need to be validate in respect to the obtained results.

Further the methodology need to be validate in accordance with the used Algorithm Optimal defense strategy selection algorithm based on attack defense game.

Additionally, I would advise the authors to verify their work for consistency in grammatical usage, formatting, and reference style

Need to relook the Efficiency of our Method section in accordance with the obtained results.

Need to relook the result and analysis section and correlate with the proposed research methodology.

Need to relook on English, figure quality, equations and connectivity among all sections.

The overall manuscript should be checked for typos, syntax, and grammar to improve the quality of content flow and presentation

The overall manuscript should be checked for typos, syntax, and grammar to improve the quality of content flow and presentation

Rest, everything is looking good and have a good technical merit of contribution with following key observation.

Originality of paper is looking good.

Technical merit of paper is good.

Include a brief statement about possible applications that could profit from the suggested methodology in the conclusions section.

Reviewer #2: 1. Briefly describe any existing models and highlight what qualities distinct your proposed NAADBG model. What are the key innovations or improvements your model offers?

2. Given that the model assumes all participants aim to optimize their outcomes, how does the model perform if attackers fail to achieve their optimal outcomes, especially in scenarios involving asymmetric information?

3. Nash Equilibrium has its limitations, such as the presence of multiple equilibria, the possibility of inefficient outcomes, and assumptions about rationality and static interactions. Please further address the limitations of your model.

4. Briefly describe any potential challenges of integrating this new model into existing security infrastructure and compatibilities.

6. PLOS authors have the option to publish the peer review history of their article (what does this mean?). If published, this will include your full peer review and any attached files.

Reviewer #1: No

Reviewer #2: No

---

## [Author Response · Author response to Decision Letter 1]

21 Oct 2024

All response has been made in the file 'Response to Reviewers' with revised manuscript.

---

## [Decision Letter · Decision Letter 1]

6 Dec 2024

A Bayesian Game Approach for Node-Based Attribution Defense Against Asymmetric Information Attacks in IoT Networks

PONE-D-24-27183R1

Dear Dr. Liu,

We’re pleased to inform you that your manuscript has been judged scientifically suitable for publication and will be formally accepted for publication once it meets all outstanding technical requirements.

Kind regards,

Brij Bhooshan Gupta

Academic Editor

PLOS ONE

Additional Editor Comments (optional):

Reviewers' comments:

Reviewer's Responses to Questions

**Comments to the Author**

1. If the authors have adequately addressed your comments raised in a previous round of review and you feel that this manuscript is now acceptable for publication, you may indicate that here to bypass the “Comments to the Author” section, enter your conflict of interest statement in the “Confidential to Editor” section, and submit your "Accept" recommendation.

Reviewer #1: All comments have been addressed

Reviewer #2: (No Response)

2. Is the manuscript technically sound, and do the data support the conclusions?

Reviewer #1: Yes

Reviewer #2: Yes

3. Has the statistical analysis been performed appropriately and rigorously? 

Reviewer #1: Yes

Reviewer #2: N/A

4. Have the authors made all data underlying the findings in their manuscript fully available?

Reviewer #1: (No Response)

Reviewer #2: Yes

5. Is the manuscript presented in an intelligible fashion and written in standard English?

Reviewer #1: Yes

Reviewer #2: Yes

6. Review Comments to the Author

Reviewer #1: The author has incorporated all suggestions in the paper, titled : A Bayesian Game Approach for Node-Based Attribution Defense Against Asymmetric Information Attacks in IoT Networks”;

Originality of paper is now looking much better.

Technical merit of paper is now looking good.

The overall manuscript have a good qualitative work.

The overall manuscript is now have a proper explanation of every definition and improved the typos, syntax, and grammar, to improve the quality of content flow and presentation

Reviewer #2: (No Response)

7. PLOS authors have the option to publish the peer review history of their article (what does this mean?). If published, this will include your full peer review and any attached files.

Reviewer #1: No

Reviewer #2: No

---

## [Editor Report · Acceptance letter]

PONE-D-24-27183R1

PLOS ONE

Dear Dr. Liu,

I'm pleased to inform you that your manuscript has been deemed suitable for publication in PLOS ONE. Congratulations! Your manuscript is now being handed over to our production team.

Kind regards,

on behalf of

Dr. Brij Bhooshan Gupta

Academic Editor

PLOS ONE